# An EGCG Derivative in Combination with Nimotuzumab for the Treatment of Wild-Type EGFR NSCLC

**DOI:** 10.3390/ijms241814012

**Published:** 2023-09-13

**Authors:** Yanping Huang, Xiangdan Cuan, Weiwei Zhu, Xingying Yang, Yunli Zhao, Jun Sheng, Chengting Zi, Xuanjun Wang

**Affiliations:** 1Key Laboratory of Pu-er Tea Science, Ministry of Education, Yunnan Agricultural University, Kunming 650201, China; fengzhongjiaren00@163.com (Y.H.); cuanxiangdan@163.com (X.C.); w52003959@163.com (W.Z.); yangxy950823@163.com (X.Y.); zhaoyunli@ynu.edu.cn (Y.Z.); shengj@ynau.edu.cn (J.S.); 2College of Science, Yunnan Agricultural University, Kunming 650201, China; 3College of Food Science and Technology, Yunnan Agricultural University, Kunming 650201, China; 4State Key Laboratory for Conservation and Utilization of Bio-Resources in Yunnan, Kunming 650201, China

**Keywords:** nimotuzumab, theasinensin A, wild-type EGFR, NSCLC

## Abstract

Inhibiting the tyrosine kinase activity of epidermal growth factor receptor (EGFR) using small-molecule tyrosine kinase inhibitors (TKIs) or monoclonal antibodies is often ineffective in treating cancers harboring wild-type EGFR. Given the fact that EGFR possesses a kinase-independent pro-survival function, more effective inhibition of EGFR-mediated signals is therefore necessary. In this study, we investigated the effects of using a combination of low-dose nimotuzumab and theasinensin A to evaluate whether the inhibitory effect of nimotuzumab on NCI-H441 cancer cells was enhanced. Here, theasinensin A, a novel epigallocatechin-3-gallate (EGCG) derivative, was identified and its potent anticancer activity against wild-type EGFR NSCLC was demonstrated in vitro; the anticancer activity was induced through degradation of EGFR. Mechanistic studies further revealed that theasinensin A bound directly to the EGFR extracellular domain, which decreased interaction with its ligand EGF in combination with nimotuzumab. Theasinensin A significantly promoted EGFR degradation and repressed downstream survival pathways in combination with nimotuzumab. Meanwhile, treatment with theasinensin A and nimotuzumab prevented xenograft growth, whereas the single agents had limited effect. Thus, the combination therapy of theasinensin A with nimotuzumab is a powerful candidate for treatment of wild-type EGFR cancers.

## 1. Introduction

Lung cancer is a leading cause of cancer-related mortality. The major pathological type, non-small cell lung cancer (NSCLC), which accounts for 70–80% of cases of lung cancer [1,2], is associated with extremely poor prognosis. The 5-year survival rate of NSCLC remains relatively low (<10%), which is partially attributed to the advanced stage of disease at the time of diagnosis and the early occurrence of distant metastases [3]. Activation of receptor tyrosine kinases has been shown to be critical in lung carcinogenesis, which has led to the development of specific targeted therapies [4]. More than 60% of NSCLC tumors overexpress epidermal growth factor receptor (EGFR), which is correlated with tumor malignancy and poor prognosis [2]. Currently, no approved small-molecule inhibitors of EGFR are available for clinical use. Over recent years, a number of strategies have been developed to inhibit the aberrant EGFR-associated signal transduction cascade [5,6]. However, in unselected patient populations treatment success with the established wild-type EGFR inhibitor gefitinib remains poor [7]. Clinical studies using chemotherapy as the first-line treatment for wild-type EGFR NSCLC indicate no significant survival advantage [8]. The development of novel anti-tumor drugs targeting wild-type EGFR with improved pharmaceutical profiles and reduced toxicity thus remains an unmet medical need.

Nimotuzumab is a humanized mAb that binds extracellular domain III of the EGFR and inhibits EGF binding [5]. Nimotuzumab has been approved in several countries for clinical treatment of various tumor types due to the absence of severe adverse effects, which is in contrast to numerous other drugs that target EGFR [9]. The low toxicity of nimotuzumab lies in its intermediate affinity, which does not interfere with the basal level of EGFR signaling [5]. Although the benefits of molecularly targeted drugs are substantial, recent research suggests that nimotuzumab partially permits kinase activity and downstream signaling under the conditions of inhibition of EGF binding [10]. Consequently, the ability of nimotuzumab to counteract growth is limited due to EGFR’s kinase-independent activity, which promotes cancer cell survival. Small-molecule compounds directly targeting EGFR and inducing its degradation could therefore present a more effective strategy to suppress wild-type EGFR activity in NSCLC and enhance therapeutic effects.

Recent research has focused on the significant utility of natural products in drug discovery and development. Compounds obtained from natural products or derivatives represent 62% of all small-molecule drugs that were approved by the U. S. Food and Drug Administration (FDA) from 1981 to 2014 [11,12]. Epigallocatechin-3-gallate (EGCG), the predominant and most active catechin of green tea, is known to have therapeutic properties in many systems. However, its effectiveness in vivo is limited owing to its poor stability [13]. EGCG is unstable as a result of the modification of hydroxyl groups, leading to reduced biological activity [14]. Here, we selected EGCG as a lead compound, synthesized a number of derivatives of it and then explored the mechanisms of action of the derivatives in wild-type EGFR NSCLC.

In the present study, we synthesized four novel derivatives of EGCG. Theasinensin A was identified as the most potent derivative; our results showed that combination of theasinensin A and nimotuzumab was a promising therapeutic strategy for NSCLCs with wild-type EGFR, both in vitro and in vivo. The effectiveness of nimotuzumab was markedly enhanced upon combining with theasinensin A. Further investigation of the range of effects and preliminary pharmacokinetic properties of theasinensin A is required to ascertain its suitability for potential application in the clinic.

## 2. Results

### 2.1. Preparation of EGCG Oxides

(−)-Epigallocatechin-3-gallate (EGCG) is one of the most abundant and biologically active molecules in green tea; it acts as a tyrosine kinase inhibitor towards cancer cells overexpressing EGFR. However, EGCG has several limitations, such as easy oxidation and easy hydrolysis by bacterial and possibly host esterases. To overcome these problems, many EGCG derivatives have been designed and synthesized. Taking into account the high oxidizability and instability of EGCG, we proposed that EGCG oxides may have greater potential to regulate EGFR signaling than EGCG. In addition, we further investigated the potential binding between EGFR and EGCG using molecular docking (Appendix A). As shown in Appendix A, it can be easily seen that the number of hydroxyl groups in the skeleton of EGCG is important, forming eight H-bonds between EGCG and six residues of EGFR (Appendix A). EGCG oxides, as polyphenols including more hydroxyl groups than EGCG, could bind to the A and B chains of EGFR.

The synthesis of EGCG derivatives (EGCG oxides) was performed according to the reaction pathways illustrated in Appendix A. EGCG was used as the starting material. EGCG was reacted with potassium hexacyanoferrate and sodium hydrogen carbonate (K_3_[Fe(CN)_6_]/NaHCO_3_) [15]; the pH was adjusted to 2 using saturated citric acid solution to yield the EGCG oxides 1 and 2 in yields of 3.2% and 5.6%, respectively. EGCG was dissolved in 30% dioxane–water and CuSO_4_·5H_2_O was added at 25 °C for 4 h to give EGCG oxide 3 in a 10.9% yield [16,17]. EGCG was reacted with phosphate buffer (Na_2_HPO_4_ [18,19], 0.05 M) at 60 °C for 2 h to obtain EGCG oxide 4 in a 90% yield. The derivatives (Appendix A) were characterized by ^1^H-NMR, ^13^C-NMR and ESI-MS; the results were consistent with the proposed structures (Appendix A).

### 2.2. Theasinensin A Directly Binds to the EGFR Extracellular Domain

To measure a possible direct interaction between EGCG derivatives and EGFR, we performed molecular interaction assays. For four EGCG derivatives at concentrations of 10 μM, the response reached 2.3, 0.2, 2.9 and −1.6 RU, respectively, indicating that compound 3 (theasinensin A) could have a relatively direct interaction with the extracellular region of EGFR (Figure 1A). The chemical structure of theasinensin A is presented in Figure 1B. To examine whether theasinensin A could directly bind to the extracellular domain of EGFR, we performed SPR on a Biacore S200 instrument, which is a novel and straight forward methodology for assessing protein–compound interactions. As shown in Figure 1C, the specific binding of theasinensin A to EGFR was detected. The kinetic data revealed that theasinensin A bound to EGFR with response values (*K*_D_, equilibrium dissociation constant) of 7.142 μM. The *K*_D_ value obtained for nimotuzumab (10.56 nM) is similar to the value reported previously (Figure 1D) [20]. However, theasinensin A (10 µM) did not impact the affinity of nimotuzumab for binding to EGFR, as the *K*_D_ was not significantly altered (19.86 nM) (Figure 1E), suggesting that nimotuzumab and theasinensin A have different EGFR binding sites. Furthermore, EGF bound to EGFR with a *K*_D_ of 0.255 µM (Figure 1F); expectedly, the addition of the nimotuzumab (50 nM) reduced the response values of EGF to EGFR (*K*_D_ of 1.406 µM) (Figure 1G). Nimotuzumab was observed to efficiently inhibit the binding of EGF [20]. An amount of 10 µM theasinensin A did not influence the affinity of EGF binding to EGFR (*K*_D_ of 0.545 µM) (Figure 1H). Importantly, the presence of both theasinensin A (10 µM) and nimotuzumab (50 nM) together did significantly impact the interaction between the extracellular domain of EGFR and EGF, increasing the *K*_D_ to 10.84 µM (Figure 1I). Taken together, these results suggest that the combination of theasinensin A and nimotuzumab directly binds to the extracellular domain of EGFR and reduces its interaction with EGF.

To further clarify the interactions between theasinensin A and EGFR, more detailed analysis was performed using molecular docking. Figure 1J illustrates the most probable theasinensin A–EGFR binding model. The key residues in the active site of EGFR, including Lys-237 and Leu-245, formed two hydrophobic interactions and Asp-238, Thr-238, Lys-229, Asn-256, Glu-259, Thr-239, Asp-238 and Lys-237 formed ten hydrogen bonds. All of the critical hydrogen bonds between theasinensin A and EGFR are presented in Appendix A. These extensive hydrogen bond interactions are suggested to play a key role in the increased binding affinity of theasinensin A for EGFR. The best binding energy of theasinensin A–EGFR was −2.23 kJ/mol. In addition, we found that nimotuzumab bound to EGFR at the structural EGF binding site (Figure 1K), further confirming that nimotuzumab can compete with EGF for EGFR binding.

### 2.3. Combination Treatment with Theasinensin A and Nimotuzumab Exhibits Potent Antiproliferative Activity in Wild-Type-EGFR-Expressing Cell Lines and Induces Apoptosis

We first evaluated the effect of the two-drug combination on four cell lines that had different EGFR mutational statuses. One cell line with the EGFRL858R/T790M double mutation (NCI-H1975 cells) and three cell lines with wild-type EGFR (A431, NCI-H441 and NCI-H1781) were chosen and their cell viability assessed via MTT assay.

Theasinensin A significantly reduced the survival of the cell lines in a dose-dependent manner, with IC_50_ values of 32 µM, 33 µM, 30 µM and 20 µM for the NCI-H441, NCI-H1781, A431 and NCI-H1975 cell lines at 48 h, respectively (Figure 2A). Similarly, nimotuzumab had a dose-dependent cytotoxic effect on the four cell lines (Figure 2B). From these results, we decided to use low-dose nimotuzumab (50 nM) in combination with theasinensin A (20 µM) to assess the synergistic effects of this combination on the viabilities of the four cell lines. Indeed, the viability was also affected by nimotuzumab, theasinensin A and their combination in NSCLC cells, whereas a mild effect was only observed for NCI-H1975 cells. In contrast, the anti-growth activity of theasinensin A in combination with nimotuzumab on wild-type EGFR in NCI-H441 cells was much stronger in the wild-type EGFR NSCLC (Figure 2C); therefore, this was selected as a model for investigation of the anti-cancer activity and the underlying mechanisms of the combination with theasinensin A. Next, data from a clonogenic assay further showed that co-treatment with theasinensin A and nimotuzumab induced significant inhibition of NCI-H441 colony formation (Figure 2D,E). In addition, flow cytometry analysis indicated that theasinensin A or nimotuzumab alone hardly caused any cell apoptosis; however, their combination dramatically promoted cell apoptosis at 48 h (Figure 2F,G). We also determined the expression levels of Bax and Bcl-2, which are hallmarks of apoptosis and play crucial roles in this cellular process. Compared with the untreated control and the individual drugs, theasinensin A plus nimotuzumab induced a significant increase in Bax and a decrease in Bcl-2. Additionally, the expression levels of p27, p53, cleaved PARP, cleaved caspase 9 and caspase 3 proteins were increased upon treatment with a combination of theasinensin A and nimotuzumab (Figure 2H,I). Taken together, these experiments show that theasinensin A plus nimotuzumab exhibits antiproliferative activity and promotes the apoptosis of NCI-H441 cells.

### 2.4. Rapid Inhibition of EGFR Phosphorylation by Theasinensin A in Combination with Nimotuzumab

To elucidate the molecular mechanisms of action of theasinensin A in conjunction with nimotuzumab in wild-type EGFR NSCLC, we examined whether drug-mediated modulation of EGFR signaling, including activation of STAT3 and ERK, is implicated in progression of NSCLC. The combination of theasinensin A and nimotuzumab induced a decrease in the levels of phosphorylated STAT3, ERK, GSK3β and EGFR, supporting a global effect on EGFR signaling (Figure 3A,B).

### 2.5. Theasinensin A Combined with Nimotuzumab Enhanced Internalization and Decreased EGFR Expression in NCI-H441 Cells

According to many studies, anti-EGFR mAbs induce internalization of EGFR, leading to downregulation of its expression on the cell surface [21,22]. Therefore, we assessed cell surface expression of EGFR in NCI-H441 cells after treatment with theasinensin A, nimotuzumab and nimotuzumab + theasinensin A to further elucidate the mechanisms involved in the combined effects. As expected, treatment of the cells with theasinensin A for 4 h resulted in extensive internalization of the tagged EGFR into vesicles located beneath the plasma membrane, which became more obvious by 7 h of treatment. However, following treatment with the combination of theasinensin A and nimotuzumab, EGFR mostly remained in the cytoplasm and was significantly less intense at 3 h than at 15 min (Figure 3C). To verify the results obtained with the immunofluorescence assay, we also assessed levels of cell-surface-associated EGFR by Western blot. Western blot analysis showed that the total EGFR protein and phosphorylated EGFR protein levels were both further decreased by theasinensin A alone or combination therapy. In the presence of EGF, the downregulation of the EGFR protein caused by nimotuzumab in combination with theasinensin A was greater than that caused by theasinensin A alone (Figure 3D). These results are consistent with our immunofluorescence studies described above. These results demonstrate that theasinensin A and nimotuzumab exhibit synergistic anti-tumor effects on NCI-H441 cells by reducing EGFR on the cell surface. Moreover, theasinensin A and nimotuzumab enhance the inhibitory effect of nimotuzumab on NCI-H441 cells. We next performed flow cytometry analysis using an anti-EGFR antibody to examine the effects on the degradation of the EGFR of treating NCI-H441 cells with theasinensin A or nimotuzumab. The median fluorescence intensity (MFI) of EGFR at 4 h was 89.99% compared with at 0 h and was subsequently maintained at 49.94% after 24 h of theasinensin A treatment in NCI-H441 cells. Taken together, these results suggest that synergy of theasinensin A and nimotuzumab was induced through the degeneration of EGFR on the surface caused by theasinensin A, resulting in an intensification of the efficacy of nimotuzumab (Figure 3E).

Since we found that theasinensin A can cause the endocytosis of EGFR in the NCI-H441 cell line, we next examined whether theasinensin A induces the ubiquitination of EGFR. Theasinensin A treatment enhanced ubiquitylation of EGFR in the presence of MG132. In addition, EGF-induced ubiquitination of EGFR was increased in the presence of theasinensin A alone or when combined with nimotuzumab when compared with the level of inhibition with nimotuzumab alone. Western blot analysis of lysates showed consistency with the hypothesis demonstrated in Figure 3F. Taken together, these findings indicate that, in contrast to the effects seen with EGF, the internalization of EGFR by endocytosis that is induced by theasinensin A is associated with ubiquitin-mediated degradation of the EGFR.

### 2.6. Treatment with Theasinensin A and Nimotuzumab Decreases NCI-H441 Tumor Growth

To confirm the superior activity against wild-type EGFR NSCLC of combining theasinensin A with nimotuzumab, in vivo experiments were performed. Specifically, nude mice subcutaneously injected with wild-type-EGFR-expressing NCI-H441 cells were administered theasinensin A and nimotuzumab. There was no significant change in the weight of the mice (Figure 4A). Although theasinensin A as a single agent moderately inhibited tumor growth, the difference was not significant compared with the control. However, the combined treatment using theasinensin A and nimotuzumab significantly suppressed tumor growth relative to the control (*p* < 0.05) (Figure 4B). Tumors were harvested at the end of the treatment period (Figure 4C); the combination therapy induced a significant decrease in tumor weight (Figure 4D). Additionally, Western blot analysis of tumor tissues demonstrated that theasinensin A plus nimotuzumab resulted in a greater suppression of p-EGFR, p-ERK1/2, p-STAT3 and p-GSK3β. Meanwhile, the combination of theasinensin A with nimotuzumab synergistically upregulated cleaved PARP and Bax expression and downregulated Bcl-2 expression in tumors. The results implied that the combination of theasinensin A with nimotuzumab vastly inhibited the tumor growth potential of NCI-H441 cells. Moreover, the efficacy of treatment is dependent on the inhibition of the phosphorylation of EGFR and its downstream signaling pathway (Figure 4E).

### 2.7. Nimotuzumab and Theasinensin A Target Phosphorylation of EGFR and ERK and Suppress Ki67 Expression

To study the therapeutic potential of a treatment combining theasinensin A and nimotuzumab, we next detected the expression of several regulatory proteins for proliferation (EGFR, ERK1/2 and Ki67) by immunohistochemical staining. As shown in Figure 5, low expression of p-EGFR, p-ERK and Ki67 were found in wild-type EGFR NSCLC samples after co-treatment with theasinensin A and nimotuzumab, whereas the total protein levels of EGFR and ERK1/2 remained unchanged in each of the groups (Figure 5A–D). The EGFR participates in the key processes of tumor cell invasion and tumor-related angiogenesis and its upregulation correlates with poor prognosis in several human tumor types [23]. Therefore, the endothelial markers CD31 and CD34 and vascular endothelial growth factor (VEGF) were used in the immunohistochemical detection of angiogenic activity in NSCLC [24]. CD31 and CD34 have been used to highlight the density of intratumorous vessels as a direct marker of the degree of neoangiogenesis. The role of VEGF in tumor angiogenesis has been extensively studied and has been shown to be a key mediator of angiogenesis in cancer, in which it is upregulated by oncogene expression and a variety of growth factors [25,26].

To observe the microvascular expression of tumor cells, the expression of VEGF, CD31 and CD34 in harvested tumor tissues was measured via IHC. Figure 4A shows that the expression of VEGF, CD31 and CD34 was moderately different between the nimotuzumab and control groups. However, the combination of theasinensin A and nimotuzumab markedly downregulated the expression of these angiogenesis-related regulators (Figure 5E,F). The combination of theasinensin A and nimotuzumab has a remarkable inhibitory effect on NSCLC tumor growth by means of inhibiting the EGFR signaling pathways and the tumor microenvironment.

## 3. Discussion

NSCLC accounts for about 85% of malignant lung tumors and is a leading cause of cancer-related mortality. EGFR, the first of four members of the ErbB family, is overexpressed in 40–80% of NSCLC cancers. Overexpression and mutation of EGFR leads to constitutive activation of the EGF/EGFR pathway, which is associated with increased tumor proliferation and chemotherapy resistance [27,28] and remains a major clinical challenge for cancer therapy. EGFR TKIs have been shown to be clinically effective in NSCLC patients with EGFR oncogene mutations. Additionally, anti-EGFR mAbs are under exploration as monotherapies and in combination with radiation, chemotherapy or other biologically targeted agents for treatment of NSCLC [29]. Nimotuzumab is a humanized IgG1 mAb that binds to extracellular domain III of EGFR with moderate affinity. Antitumor activity with minimal skin toxicity for nimotuzumab relative to other anti-EGFR drugs has been reported for specific cancer types. Further research has validated the utility of nimotuzumab in combination with thoracic radiation for NSCLC [30]. Moreover, research has demonstrated the potential to develop other mAb nanoparticle complexes for a superior therapeutic efficacy. AuNP-NmAb has great potential in the treatment of EGFR+ cancers [31,32]. This prompted us to speculate that nimotuzumab combined with novel natural compounds could exert robust clinical effects with manageable safety profiles as potential therapies targeting EGFR in NSCLC. In this study, we focused on the efficacy of a combination treatment using theasinensin A and nimotuzumab against EGFR-overexpressing NSCLC, both in vitro and in vivo.

The potential biological activity of epigallocatechin gallate (EGCG, (−)-epigallactechin-3-gallate) has been extensively researched, including its anti-cancer, antioxidant and anti-inflammatory effects [33,34,35]. However, EGCG has a number of limitations, such as low solubility and stability. To improve these characteristics, several investigators have developed EGCG derivatives and generated chemically modified EGCG [36]. In vitro molecular interaction assays have revealed relatively direct interactions between the novel EGCG derivative theasinensin A and the EGFR extracellular fragment. We hypothesized that theasinensin A influences nimotuzumab binding to EGFR, exerting a range of biological effects. The combination of these two drugs inhibited the cell viability of EGFR-overexpressing NSCLC cells, which was not evident in the double-mutant EGFR 1975 cell line, clearly suggesting a synergistic effect of theasinensin A and nimotuzumab in EGFR-expressing NSCLCs. EGCG and EGFR-TKI induce apoptosis in a variety of cancer types, including NSCLC, SCCHN and TNBC [37,38,39,40,41]. In cells co-treated with theasinensin A and nimotuzumab, NCI-H441 cell apoptosis was also investigated using annexin V–FITC and PI fluorescence staining. It was notable that theasinensin A plus nimotuzumab treatment led to a significantly increased expression ratio of Bax/Bcl-2, which further supports that theasinensin A plus nimotuzumab was capable of inducing apoptosis in NCI-H441 cells and that theasinensin A induces apoptosis via Bcl-2 and Bax modulation. These results suggest that the effect of combined treatment on cell growth inhibition in NSCLC is at least partially attributable to apoptosis.

Numerous studies have suggested that both EGCG and erlotinib significantly inhibit p-EGFR, p-ERK and p-AKT in SCCHN and that this is associated with a marked increase in apoptosis [38,40,41]. Similar experimental results were obtained for NSCLC [37]. Consistent with these observations, the combined treatment with theasinensin A and nimotuzumab synergistically inhibited phosphorylation of EGFR and downstream signaling. Our findings indicate that the endocytosis and turnover rate of EGFR are also potential molecular determinants. Internalization of membrane EGFR may disrupt EGFR-specific signaling, presenting a mechanism that may underlie the synergistic inhibition of tumor cell growth. With the aid of parallel immunofluorescence and Western blot analyses, we demonstrated rapid degradation of EGFR by theasinensin A, both alone and in the presence of EGF. Further experiments revealed that the EGFR internalized by theasinensin A underwent ubiquitin-dependent degradation, limiting the amount of EGFR that could be recycled back to the membrane. Treatment with the proteasomal inhibitor MG132 and theasinensin A induced an increase in ubiquitinated EGFR along with the accumulation of total EGFR protein. Stabilization of EGFR at the membrane by nimotuzumab may facilitate capture of either phosphorylated or inactivated EGFR by theasinensin A, resulting in the suppression of EGFR signaling. The link between DNA repair mechanisms and epidermal growth factor receptor (EGFR) signaling has been reported in many human tumor cells [42]. Hence, additional studies are necessary to understand the complex EGFR signaling pathway induced by oxidative stress factors.

Although the potency of theasinensin A monotherapy was not completely satisfactory, combination with an anti-EGFR antibody significantly reduced the IC_50_ value obtained against NSCLC cells overexpressing EGFR by inducing degradation of surface EGFR expression and leading to tumor shrinkage in NCI-H441-xenograft-bearing mice. Most importantly, compared with control and single agent treatment groups, the combined treatment resulted in greater inhibition of growth (assessed using Ki67) and the angiogenic factors CD34, CD31 and VEGF; this was mediated via significant suppression of the p-EGFR and p-AKT pathways.

## 4. Materials and Methods

### 4.1. Cell Lines and Reagents

All cell lines were purchased from Kunming Institute of Zoology, Chinese Academy of Sciences (Kunming, China). NCI-H1975, NCI-H441, NCI-H1781 and A431 cells were cultured in RPMI-1640 and DMEM high-glucose medium (Thermo Fisher Scientific, Pittsburgh, PA, USA) supplemented with 50 IU/mL penicillin, 50 mg/L streptomycin (Solarbio, Beijing, China) and 10% fetal bovine serum (HyClone, Los Angeles, CA, USA) at 37 °C in a humidified 5% CO_2_ incubator. The synthesis of the EGCG derivatives was shown in Appendix A.

### 4.2. Molecular Interaction Assay

The binding affinity of the EGCG derivatives for EGFR was determined with an Octet Red96 system (Pall Fort’eBio, Fremont, CA, USA). The biotinylated EGFR protein was loaded on Super Streptavidin biosensors and activated with gradient concentrations of EGCG derivatives in the assay buffer (PBS, pH 6.5). We measured EGCG derivative association and dissociation for 300 s each. The kinetic parameters and affinities were calculated from a nonlinear global fit of the data between EGCG derivatives and EGFR using Octet Data Analysis v.7.0 (Pall Fort’eBio, Fremont, CA, USA).

Measurement of the binding affinities was performed using surface plasmon resonance (SPR, BIAcore S-200, Chicago, IL, USA). EGFR was coupled to a CM5 BIAcore sensor using standard amine coupling. Optimal coupling was obtained in 10 mM sodium acetate at pH 4.5 for EGFR. The kinetics and affinity assay were determined at a flow rate of 30 mL/min using HBS-EP8 (10 mM HEPES buffer, 150 mM NaCl, 3 mM EDTA and 0.005% Tween 20, pH 8.0) buffer. A reference flow cell without protein was used for background subtraction. Theasinensin A was double-diluted in PBS-P buffer (GE Healthcare, Chicago, IL, USA) supplemented with 2% DMSO to concentrations ranging from 3.125 to 50 μM. Different concentrations of compounds were injected at a rate of 30 µL/min. After each measurement, the association and dissociation times were both 60 s and the surface was regenerated with 2 × 30 s injections of 1 mol/L glycine (pH 2.5) and 1 mol/L NaCl. The binding kinetics of theasinensin A to EGFR were analyzed using Biacore S200 Evaluation Software Version 1.1 (GE Healthcare, Chicago, IL, USA) via a 1:1 binding model. The affinity values were calculated from *k*_d_/*k*_a_ ratios, where *k*_a_ is the association rate constant (M^−1^s^−1)^ and *k*_d_ is the dissociation rate constant (s^−1^). Data were analyzed using Prism 5 (GraphPad Software, Inc. San Diego, CA, USA).

### 4.3. Cell Viability Assay

NCI-H441 cells were seeded into 96-well plates (3 × 10^4^) and treated with EGCG derivatives (20 µM), nimotuzumab (50 nM) or nimotuzumab (50 nM) combined with EGCG derivatives (20 µM) for 48 h before being subject to the MTT assay. Cell viability was determined by measuring the absorbance at 492 nm using a FlexStation 3 Multi-Mode Microplate Reader (Molecular Devices, Sunnyvale, CA, USA).

### 4.4. Colony Formation Assay

NCI-H441 cells were seeded into 60 mm plates (5000 cells/well) and incubated overnight. Then, theasinensin A (20 µM) and/or nimotuzumab (50 nM) was added to each plate and the culture medium was changed every 3 days. After 14 days of incubation, the cells were fixed in 4% paraformaldehyde and then stained with a 1% crystal violet solution.

### 4.5. Cell Apoptosis Assays

NCI-H441 cells were seeded in 6-well plates at a density of 5 × 10^5^ cells per well, harvested and washed twice with PBS. The cells were treated with theasinensin A (20 µM) with or without nimotuzumab (50 nM) in serum-free medium for 24 h. The harvested cells were then incubated with 100 µL of binding buffer containing 5 µL of Annexin V–FITC and 5 µL of PI (20 µg/mL) in the dark for 15 min at room temperature. Subsequently, the prepared samples were analyzed using BD FACSCalibur flow cytometry within 1 h and the percentage of apoptotic cells was determined using FlowJo V10 software.

### 4.6. Flow Cytometry Analysis

NCI-H441 cells (5 × 10^5^) were seeded into 6-well plates. After treatment, the cells were fixed in 4% formaldehyde for 10 min and incubated with primary antibody for 1 h at room temperature. The primary antibody used in the study was rabbit monoclonal anti-EGFR (5616, Cell Signaling Technology, Boston, MA, USA) at a dilution of 1:50. Data were obtained with a FACScan Flow cytometer (BD Biosciences, New York, NJ, USA) by collecting a minimum of 10,000 events and analyzed using FlowJo software. The data were expressed as fluorescence mean intensity (FMI).

### 4.7. Antibodies and Western Blotting

The total proteins were first separated by 8–10% sodium dodecyl sulfate-polyacrylamide gel electrophoresis (SDS–PAGE) and then electrophoretically transferred onto supported nitrocellulose membranes. Next, they were blocked with 5% skimmed milk in PBST for 1 h. The membranes were blocked and incubated with the indicated primary antibodies, including phospho-EGFR (Tyr1068; ab5644, 1:1000) (Abcam, Cambridge, Britain), total EGFR (sc-373746, 1:1000) (Santa Cruz Biotechnology, Santa Cruz, CA, USA), phospho-Akt (Ser473; 4060, 1:1000), total Akt (4691, 1:1000), phospho-ERK (Thr202/Tyr204; 9101, 1:1000), total ERK1/2 (9102, 1:1000) (Cell Signaling Technology) and β-actin (A5228, 1:1000) (Sigma-Aldrich, Shanghai, China) antibodies, followed by incubation with the corresponding secondary antibodies (HAF007, HAF008; 1:5000) (R&D Systems, Minneapolis, MN, USA). An ultra-sensitive enhanced chemiluminescent substrate kit (4 A Biotech Co., Ltd., Beijing, China) was applied to detect the chemiluminescence signals. The bands of interest were captured using the FluorChem E System (ProteinSimple, San Jose, CA, USA) and quantified using AlphaView software 3.3 (Cell Biosciences, Santa Clara, CA, USA).

For immunoprecipitation studies, about 300 µg of lysate was precleared with recombinant proteinA/G–agarose (GIBCOBRL, Carlsbad, CA, USA) for 4 h at 4 °C, then incubated with 2 µg of antibody against the N-terminus of EGFR (528, SC-120, Santa Cruz, CA, USA) or nonimmunized mouse IgG pre-complexed with protein G–agarose overnight. The membrane was the probed with an anti-ubiquitin antibody (Cell Signaling Technology).

### 4.8. Immunofluorescence Staining

Cells were starved in serum-free medium for 4 h and pre-exposed to 20 µM theasinensin A, 50 nM nimotuzumab or a combination of both for 4 h, followed by 20 ng/mL EGF for 15 min. Alternatively, cells were pre-exposed to 20 µM theasinensin A, 50 nM nimotuzumab or a combination of both for 7 h, followed by 20 ng/mL EGF for 3 h. Cells were washed twice in PBS, fixed in 4% formaldehyde and permeabilized with 1% Triton X-100 at room temperature. Next, the cells were blocked in 5% bovine serum albumin (BSA) and then incubated with an EGFR antibody (sc-373746, 1:1000) (Santa Cruz Biotechnology, Santa Cruz, CA, USA) overnight at 4 °C, washed with PBS and incubated with Alexa 488-conjugated secondary antibody (4412, 1:500) (Cell Signaling Technology) for 1 h in the dark. Subsequently, the cells were washed with PBS and mounted with 4′,6-diamidino-2-phenylindole (DAPI). Fluorescent images were obtained with confocal microscopy (Leica, Frankfurt, Germany).

### 4.9. Growth of Mouse Xenograft Tumors

All mouse experiments were performed according to guidelines set by the Yunnan Agricultural University Institutional Animal Care and Use Committee—approved animal protocols. Thirty male BALB-C/nude mice (aged 6 to 8 weeks, ~25 g in weight) were purchased from Cavens Lab Animal (Changzhou, China) and allowed to acclimate for 1 week. The animals were housed in polypropylene cages with sterile paddy husks and were maintained under standard pathogen-free conditions (ambient temperature 24 ± 1 °C, humidity 50–60%, 12 h light/dark cycle) with free access to a standard laboratory diet and water. Mice were randomly divided into four groups using randomization such that the mean weight of each group was equal (control, 40 mg/kg theasinensin A or 2.5 mg/kg nimotuzumab). NCI-H441 cells (5 × 10^6^) were suspended in 200 µL buffer and subcutaneously implanted into the right back of the mice. Tumors began to appear on the fourth day after the tumor cells were injected. Three days later, the mice were intraperitoneally injected with theasinensin A and nimotuzumab (three times per week) for the indicated experimental period. The body weights of the mice were monitored twice weekly. Tumor growth was monitored thrice weekly, and relative tumor volume was calculated as 0.5 × length × width × width (mm^3^).

### 4.10. Immunohistochemistry

Paraffin-embedded sections of tumor tissues were stained with pERKT202/Y204 (4370, 1:100), pEGFR (4060, 1:500) (Cell Signaling Technology) and Ki67 (ab193363, 1:500) (Abcam) for immunohistochemistry. The sections were blocked with serum at 37 °C for 20 min and were then incubated with the primary antibody overnight at 4 °C. After rewarming at RT for 30 min, the slides were treated with a 3,3′-diaminobenzidine (DAB) substrate solution to detect pEGFR, pERKT202/Y204 and Ki67 expression and the sections were counterstained with hematoxylin. Images were captured using a CKX41 microscope (Olympus, Tokyo, Japan) at 400× magnification. Protein immunoreactivity was quantified using Image-Pro Plus software 6.0.

### 4.11. Molecular Docking Study

The Autodock 4.2 program was employed to perform docking calculations between theasinensin A and EGFR. The structure of theasinensin A was obtained from Chem 3D and the crystal structure of EGFR (PDB ID 3njp) from the Protein Data Bank (http://www.rcsb.org/, accessed on 1 January 2021). The geometry of theasinensin A and EGFR was optimized using Avogadro software (Version 1.90.0 http://avogadro.cc/, accessed on 1 January 2021). Docking was performed using the Lamarkian genetic algorithm (LGA). The number of GA runs was set to 100 and the highest populated cluster with lowest energy conformation based on scoring function selected as the binding mode. The conformation with the lowest energy was selected for visual analysis using PyMol v1.6.x.

### 4.12. Statistical Analysis

Data are shown as means ± the standard error of the mean (SEM). All experiments were performed at least three times and representative images are shown. The one-way ANOVA was performed using SPSS 19.0 (IBM, Armonk, NY, USA). A value of *p* < 0.05 was considered significant.

## 5. Conclusions

Taken together, our results, for the first time, suggest that the combination treatment of theasinensin A and nimotuzumab is active against wild-type EGFR NSCLC cells via the induction of EGFR degradation and the inhibition of downstream survival pathways. these findings will hopefully provide a prospective strategy for NSCLC patients with wild-type EGFR.

## Figures and Tables

**Figure 1 ijms-24-14012-f001:**
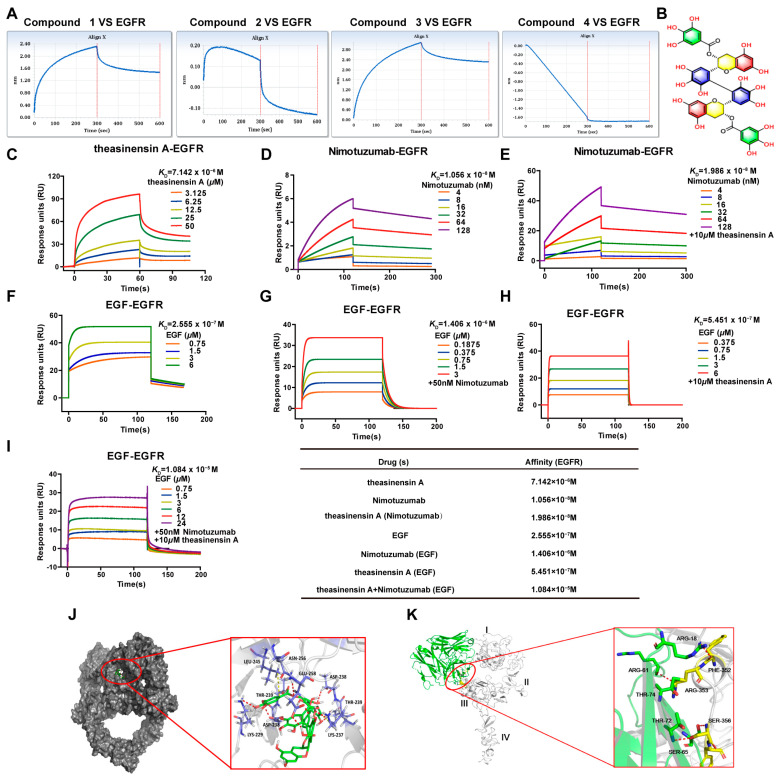
(**A**) Interactions between EGCG derivatives (10 µM) and extracellular EGFR measured using the Octet RED96 system with association and dissociation times of 300 s. (**B**) Chemical structure of theasinensin A. (**C**,**D**) Theasinensin A and nimotuzumab bind to EGFR with strong affinity, as validated by SPR analysis. (**E**) Nimotuzumab binding to EGFR in the presence of theasinensin A. (**F**) EGF binding to EGFR. (**G**) EGF binding to EGFR in the presence of nimotuzumab. (**H**) EGF binding to EGFR in the presence of theasinensin A. (**I**) EGF binding to EGFR in the presence of nimotuzumab (50 nM) and theasinensin A (10 µM). (**J**) Intermolecular interactions of theasinensin A docked to EGFR. EGFR is depicted as a surface model (grey), theasinensin A as sticks (green), individual residues as sticks (C, royal blue; N, blue; O, red; and H, hydrogen), hydrogen bonds as dashed red lines and hydrophobic interactions as yellow dotted lines. (**K**) The binding models of nimotuzumab and EGFR.

**Figure 2 ijms-24-14012-f002:**
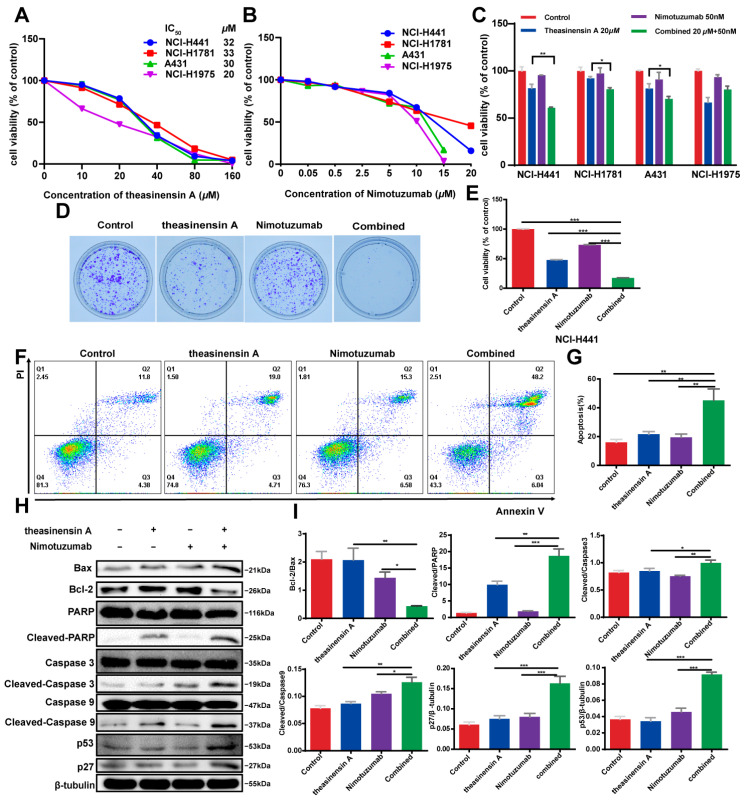
The combination treatment with theasinensin A and nimotuzumab exhibits a potent antiproliferative activity and induces apoptosis. (**A**) NCI-H441, NCI-H1781, A431 and NCI-H1975 cells were treated with 0–160 µM theasinensin A for 48 h, and survival was assessed with the MTT assay. (**B**) The indicated cells were treated with 0.05–20 µM nimotuzumab A for 48 h, and survival was assessed with the MTT assay. (**C**) The cell growth inhibition of NCI-H441, NCI-H1781, A431 and NCI-H1975 cells treated with theasinensin A (20 µM) with or without nimotuzumab (50 nM) for 48 h, as assessed by MTT assay. Statistical significance was assessed versus control. The significance level was set at *p* ≤ 0.05 (*), *p* ≤ 0.01 (**) or *p* ≤ 0.001 (***). (**D**) Effects of theasinensin A (20 µM) and nimotuzumab (50 nM) on the colony formation and cell proliferation of NCI-H441 cells. (**E**) Quantification of clonogenic formation. (**F**) Apoptosis of NCI-H441 cells was evaluated by flow cytometry. (**G**) The ratio of apoptotic cells in each group are expressed as percentages. (**H**) The expression levels of the Bcl-2/Bax, p27, p53, cleaved PARP, cleaved caspase 9 and caspase 3 proteins were determined by Western blot analysis. (**I**) Quantification of Western blot. The significance level was set at *p* ≤ 0.05 (*), *p* ≤ 0.01 (**) or *p* ≤ 0.001 (***). The results in (**A**–**C**,**E**,**G**,**I**) are expressed as mean ± SEM.

**Figure 3 ijms-24-14012-f003:**
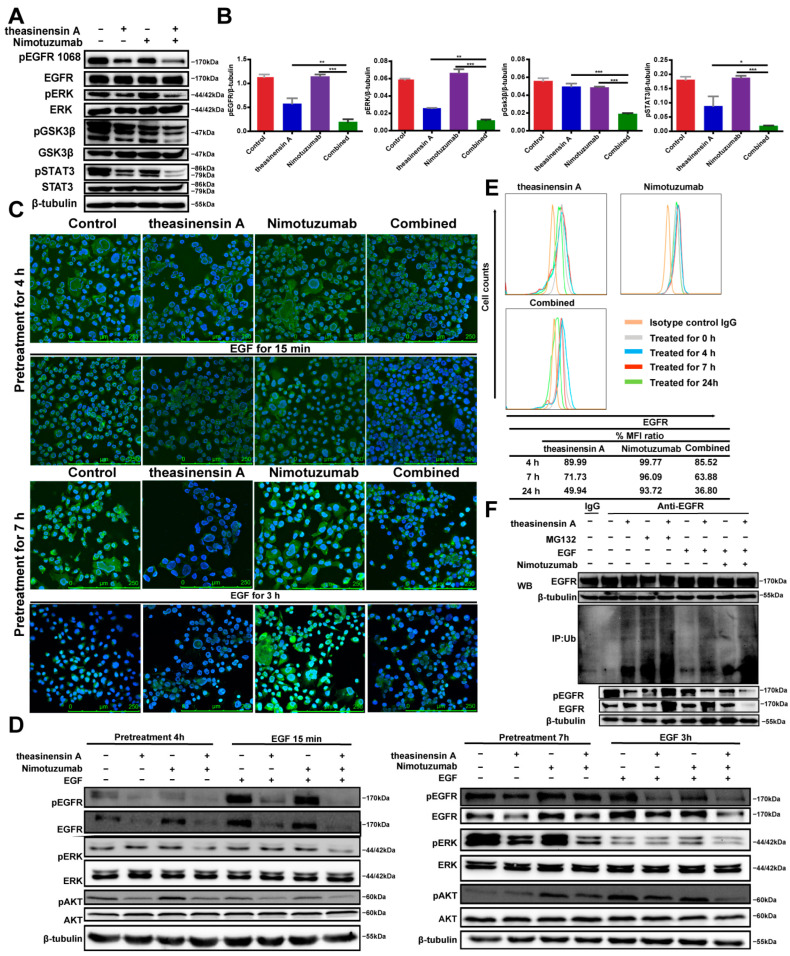
Theasinensin A combined with nimotuzumab enhanced internalization and reduced EGFR expression in NCI-H441 cells. (**A**) Inhibition of the EGFR signal pathway in NCI-H441 cells. (**B**) Quantification of Western blot. The significance level was set at *p* ≤ 0.05 (*), *p* ≤ 0.01 (**) or *p* ≤ 0.001 (***) vs. combination therapy group. (**C**) Immunofluorescence staining of EGFR (×400). (**D**) Immunoblot analysis was performed for EGFR and downstream signaling protein expression after drug treatment. (**E**) FACS analysis using a PE-conjugated EGFR antibody in NCI-H441 cells. (**F**) Theasinensin-A-induced ubiquitination of EGFR.

**Figure 4 ijms-24-14012-f004:**
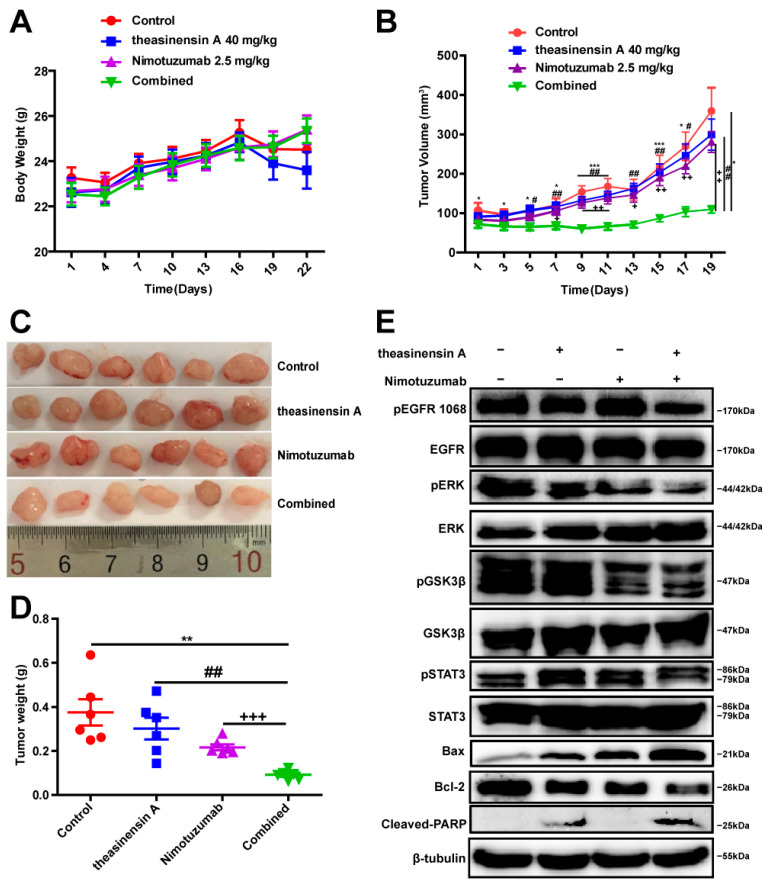
Theasinensin A in combination with nimotuzumab induces tumor regression in the NCI-H441 xenograft tumor model. (**A**) The body weights of mice were measured twice weekly. (**B**) Tumor volume was calculated as 0.5 × length × width^2^. * *p* < 0.05; *** *p* < 0.001, control vs. combination therapy group; ^#^ *p* < 0.05; ^##^ *p* < 0.01, theasinensin A vs combination therapy group; ^+^ *p* < 0.05; ^++^ *p* < 0.01, nimotuzumab vs. combination therapy group. (**C**) Images of dissected tumors from the indicated groups (**D**) and tumor mass (*n* = 6). Results are expressed as mean ± SEM, ^+++^ *p* < 0.001, ^##^ *p* < 0.01 or ** *p* < 0.01 versus indicated groups. (**E**) Western blot of NCI-H441 cells showing that co-treatment with theasinensin A and nimotuzumab suppresses phosphorylation of EGFR and downstream signaling. The levels of apoptosis-related proteins were further analyzed.

**Figure 5 ijms-24-14012-f005:**
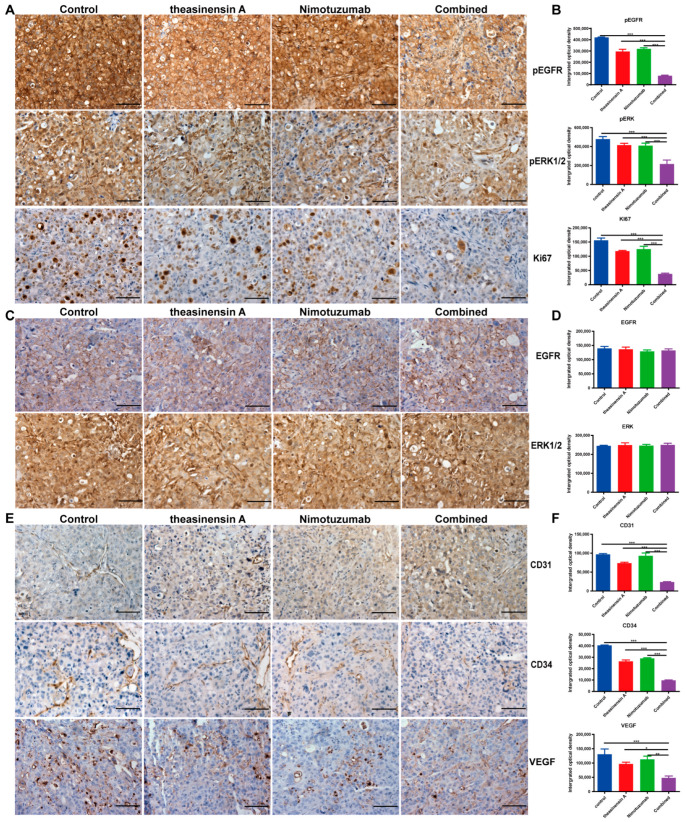
Effects of treatment with theasinensin A, nimotuzumab, and both agents in NCI-H441 xenograft tumors. (**A**) Representative image of the immunohistochemical analyses of p-EGFR, p-ERK and Ki67 in NCI-H441 tumors (scale bar = 50 µm). (**C**) Representative images of immunohistochemical analyses of EGFR and ERK in NCI-H441 tumors after three weeks of treatment (scale bar = 50 µm). (**E**) IHC analyses of CD31, CD34 and VEGF in NCI-H441 tumors after three weeks of treatment (scale bar = 50 µm). (**B**,**D**,**F**) Graphs represent means ± SEM. Expression levels were calculated based on the integrated optical density (IOD) value (*n* = 6–15 images per group, * *p* < 0.05, ** *p* < 0.01, *** *p* < 0.001).

## Data Availability

All data generated or analyzed during this study are included in this article and its Appendix A.

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
