# Peer review of "An EGCG Derivative in Combination with Nimotuzumab for the Treatment of Wild-Type EGFR NSCLC"

_ijms, 2023, doi:10.3390/ijms241814012_

Round 1

Reviewer 1 Report

Authors have reported that combination therapy of theasinensins A with nimotuzumab for the therapy of Epidermal growth factor receptor. The authors have conducted in vitro, in vivo and in silico studies. Moreover, in in vitro studies, authors performed MTT assay followed byc olony formation assay, which was later subjected to immunoblotting and flow cytometric analysis. the same was done using tumor xenograft models.

Introduction, material and methods and results and discussion sections are well written. However, abstract and conclusions sections need some attention to revise them before acceptance. Some of the finding should be added to abstract as it is more generalized and conclusions is not written properly. The current research can be published after minor revision.

Some of the more comments are as follow:

Typo errors throughout manuscript. 

formulas should be wriiten in proper format like check line 99-100.

digits/numbers should be significant and donot exceed more than 3 values after decimal, these can be rectified throughout manuscript like authors should check line 118, 120 and so on.

Reviewer 2 Report

The manuscript was aimed to examine the activity of theasinensins A, the EGCG derivate used alone and in combination with nimotuzumab (the humanized anti-EGFR monoclonal Ab) against the wild type non-small cell lung cancer (NSCLC) cells in vitro and in vivo. The manuscript is well-prepared and illustrates the novel data about the efficacy of this combination against NCI-H1441 lung cancer cells. 

I have the following concerns and suggestions about this manuscript:

1) The numbers of the Supplementary Figures should be re-numerated. For example, Fig. S3 comes first (lines 89-90), whereas Fig. S1 is referred on the line 95. 

2)  The authors have to explain the reason of examining the expression of total forms of  p53 and p27, as was referred in line 176 and  shown in the Figure 2H. The changes in the cell cycle profile was not examined in the manuscript and activation of the DNA damage response was not shown. Moreover, the expression of phosphorylated form p53 is not included in this Figure. In addition, the expression of cleaved caspase-3 usually is  composed of 2 bands (17 and 19 kDa), whereas one band only was shown in Fiogure 2H. 

3) The expression of the total and activated forms of AKT is also desirable for the Figure 3A, to provide a whole picture of the activation cascade.

4) The authors highlighting about the synergistic effects between theasinensines A and nimotuzumab (e.g., the lines 162, 222, etc.). However, the synergy score was not examined in the manuscript, and based on the data shown in Figure 2C this effect looks most likely as the additive. Thus, the performance of the Synergy Score is highly desirable to deliniate between the additive and synergistic effects of the compounds indicated above.  

5) The authors showed that the treatment of NCI-H1441 cells with theasinensins A resulted in the extensive internalization of the EGFR into the vesicles localized beneath the plasma membrane ( as shown in Figure 3C). After illustrating this data, the authors also showed the decreased expression of the total and phosphorylated EGFR proteins examined in the whole-cell lysates (WCLs) by WB. Since the experiment was done on the WCLs (the cell samples were not fractionated), the authors have to clear this point how this WB data (Figure 3C) supports the immunofluorescence data, as was shown in  Figure 3C.  

Minor:

1) Similar to the IHC-staining ( section 4.10), the cat. numbers of mAbs used for WB and IF are also recommended to show (4.7 and 4.8, respectively).

Reviewer 3 Report

it is an interesting study focusing on the role of wild-type EGFR in lung tumor progression and the effects of anti-oxidant factors along with anti-EGFR monoclonal antibodies. Please explain in the methods wether a cutoff level was established about immunohistochemistry.

Please be willing to explain how the ANOVA test was applied and timeline of the experiments.Please include the perspectives of these findings. I suggest to discuss about the EGFR signalling pathway induced by oxidative stress factors which involves a wide range of cytoplasmic and nuclear factors involved in tumor growth, such as PI3K, p16, EMT, HIF.

I suggest to include the following references to widen the discussion about nimotuzumab and the mechanisms underlying tumor progression

-J Funct Biomater. 2023 Aug 1;14(8):407. 

-Thorac Cancer. 2020 Nov;11(11):3060-3070. 

Reviewer 4 Report

This is an excellent preclinical study examining the use of theasinensins A and nimotuzumab alone and in combination to target epidermal growth factor receptor (EGFR), which has implications for the treatment of non-small cell lung cancer. The authors found that both agents are able to bind to wild-type EGFR, thereby reducing EGFR expression, inhibiting downstream signaling pathways, and causing apoptosis with synergistic effects. In vivo studies using a murine model showed decreased tumor volume as a result of combination therapy. The methods and results appear scientifically sound. Below are a few minor comments regarding the discussion and conclusion.

1. Several sentences in the discussion could benefit from elaboration and/or additional references.

For example, line 330-331: "The potential biological activity of epigallocatechin gallate (EGCG, (-)-epigallactechin-3-gallate) has been extensively researched." Please mention specifically what biologic effects have been previously investigated and provide references.

Line 340-341: "EGCG and EGFR-TKI induce apoptosis in a variety of cancer types, including NSCLC." Please mention specific other cancer types and provide references.

Line 349-350: "Numerous studies suggest that both EGCG and erlotinib significantly inhibit p-EGFR, p-ERK and p-AKT in SCCHN, associated with a marked increase in apoptosis." Here, only one study is cited although "numerous" studies are mentioned. Please include additional references.

2. Suggest rephrasing the first line of the conclusion, which is given as "In conclusion, combination therapy with nimotuzumab and theasinensins A exerts more beneficial effects than monotherapy in vitro with no toxicity." Although reported toxicity of the examined agents is low, "with no toxicity" would be an exaggeration. Also, because toxicity was not assessed in this study, would suggest rephrasing this to say "with low expected toxicity," reflecting the theoretical nature of this claim. The conclusion only mentions in vitro effects although in vivo effects were also observed in this study, which may also be worth mentioning.

Reviewer 5 Report

Author should use "Theasinensin A" instead of "Theasinensins A".

Page no.2 line 74: Author should rephrase the statement as his hypothesis, but here author stated a statement like it already a established fact. To give a clear hypothesis author should rephrase this whole paragraph, as what was the hypothesis and and what observed.

In fig 1 A : Compound 3 is missing, and at third position compound 2 is mentioned. Author should clarify this ambiguity. 

Fig 1 D and E containing same label, Author should clear this ambiguity.

Fig 1 F,G, H and I, all have same label, Author should label properly each figure as per their legend and description.

Author should maintain homogeneity in the image labelling and their legend description.

In molecular interaction assay result section author used Octet Red96 (Brand Name and Manufacturer???) instrument, but in methodology Author mentioned Surface Plasmom Resonance, Author should justify the methodology and their result.

Page No. 11 Line 379, Author should use COinstead of CO2.

Molecular interaction methodology should include a brief methodology how it perform in your laboratory as standard protocol or any modifications.

Author should use proper mathematical notations like 3 x 104 instead of 3 x 104.

In cell viability assay author should provide information of used concentration of nimotuzumab or nimotuzumab combined with EGCG derivatives.

MTT assay done by using commercial kit (Kit Details) or author prepared all required buffers and reagents in his/her lab.

Author should provide cell count used in colony formation assay.

Author should provide used concentration of theasinensin A, and/or nimotuzumab in colony formation assay.

line 405:  Author should use 10instead of 105.

Author should provide details of used concentration of theasinensin A, and/or nimotuzumab in cell apoptosis assay.

Author should provide information of seeding cell count in flow cytometry assay.

Line 448:  Author should use10instead of 106.

Author should provide details of xenograft location.

Author should provide details of experimental period in the Growth of mouse xenograft tumors section.

Author should use mm3 instead of mm3.
